# Circadian regulation of night feeding and daytime detoxification in a formidable Asian pest *Spodoptera litura*

Jiwei Zhang[1,2], Shenglong Li[1], Wanshun Li[1,2], Zhiwei Chen[1], Huizhen Guo[1,2], Jianqiu Liu[2], Yajing Xu[1,2], Yingdan Xiao[1,2], Liying Zhang[1,2], Kallare P. Arunkumar[3], Guy Smagghe[4,5], Qingyou Xia[1,2], Marian R. Goldsmith [6 ✉], Makio Takeda[7 ✉] & Kazuei Mita [1,2 ✉]

Voracious feeding, trans-continental migration and insecticide resistance make *Spodoptera litura* among the most difficult Asian agricultural pests to control. Larvae exhibit strong circadian behavior, feeding actively at night and hiding in soil during daytime. The daily pattern of larval metabolism was reversed, with higher transcription levels of genes for digestion (amylase, protease, lipase) and detoxification (CYP450s, GSTs, COEs) in daytime than at night. To investigate the control of these processes, we annotated nine essential clock genes and analyzed their transcription patterns, followed by functional analysis of their coupling using siRNA knockdown of interlocked negative feedback system core and repressor genes (*SlituClk*, *SlituBmal1* and *SlituCwo*). Based on phase relationships and overexpression in cultured cells the controlling mechanism seems to involve direct coupling of the circadian processes to E-boxes in responding promoters. Additional manipulations involving exposure to the neonicotinoid imidacloprid suggested that insecticide application must be based on chronotoxicological considerations for optimal effectiveness.

[1] State Key Laboratory of Silkworm Genome Biology, Southwest University, Chongqing, China. [2] Biological Science Research Center, Southwest University, Chongqing, China. [3] Central Muga Eri Research and Training Institute, (CMER&TI), Central Silk Board, Lahdoigarh, Jorhat, India. [4] College of Plant Protection and Academy of Agricultural Sciences, Southwest University, Chongqing, China. [5] Department of Plants and Crops, Laboratory of Agrozoology and International Joint China-Belgium Laboratory on Sustainable Control of Crop Pests, Ghent University, Ghent, Belgium. [6] Department of Biological Sciences, University of Rhode Island, Kingston, RI, USA. [7] Graduate School of Agricultural Science, Kobe University, Kobe, Japan. ✉email: mki101@uri.edu; mtakeda@kobe-u.ac.jp; mitakazuei@gmail.com

*S*podoptera litura belongs to the family Noctuidae, so-named because larvae feed and adults fly at night. This insect is called "night thief" in Japan, because the voracious feeding of the late instar larvae is intense at night, while, in contrast, they cryptically hide in the soil during the day. In addition, this species is highly polyphagous, ranging over 120 host plants (https://www.cabi.org/isc/datasheet/44520). Night feeding also determines the exposure time to plant secondary metabolites and xenobiotics such as insecticides. This dichotomous metabolic situation may explain why this species is considered a "super-pest" across Asia.

Comparative analyses using genome databases published for lepidopteran species, including *Spodoptera frugiperda*, *Helicoverpa armigera*, *Helicoverpa zea*, *Manduca sexta*, *Plutella xylostella*, *Heliconius melpomene,* and *Bombyx mori*[1–6], reveal the extraordinary ability of *S. litura* for detoxification and tolerance to many plant secondary metabolites and xenobiotic agents, including pesticides, through a great expansion of detoxification gene families and gustatory receptor genes[7]. In addition, population genetic analysis of *S. litura* uncovers the long-distance migration by which this pest expanded its territories and established genomic uniformity in Asia[7]. These properties, along with its high fecundity, make *S. litura* one of the most difficult pests to control.

A similar expansion of detoxification gene families and long-range migration occurs in the congener, the fall armyworm *S. frugiperda*. Previously restricted to the American continent, in 2016 this species was reported to have invaded Africa with outbreaks that caused extensive crop damage[8]. In the same year, it also appeared in India, expanding widely in South India[9]. The fast and extensive invasion of a highly pestiferous noctuid with similar genomic characteristics to *S. litura* highlights the urgency of disseminating new and effective control strategies to block the expansion of these highly destructive night-feeding insects.

Circadian clocks synchronize rhythms using a central time-keeper, which also controls numerous aspects of physiology and behavior, including feeding and locomotion, sleep/wake cycle, hormone and neurotransmitter secretion, and developmental events such as molting and metamorphosis[10–14]. The feeding and digestion of another congener, *S. littoralis*, are reported to be linked with circadian rhythms[15], as are larval activity and feeding in *Heliothis virescens*, another noctuid pest[16]. This may be a common feature with *S. litura*, which exhibits similar circadian processes. The expression and activity of xenobiotic-metabolizing genes which fluctuate in daily rhythms in *Drosophila melanogaster* and mosquitos[17,18], as well as in mammalian systems[19], have not yet been well-documented in Lepidoptera. Transcriptional activation, which is dependent on the two central circadian regulators, CLOCK (CLK) and CYCLE (rather, BMAL1 in Lepidoptera[20,21]), is initiated by the binding of their heterodimer to CACGTG, a canonical E-box element of target genes, including *period* (*Per*) and *timeless* (*Tim*) in *D. melanogaster*[22,23]. After *Per* and *Tim* are transcribed and translated, PER forms a heterodimer with TIM, which translocates to the nucleus, where it inhibits CLK-CYC-mediated transcription of *Per* and *Tim*[24,25]. Although some molecular partners differ among species, this basic framework of a transcription/translation-based negative feedback loop is conserved among almost all organisms examined from Archaea to humans[26–29], including Lepidoptera[15,16]. In addition to the conserved transcription–translation feedback loops, some other trans-acting factors have been discovered. For example, CWO, the gene product of *clockwork orange* (*Cwo*), a transcriptional repressor belonging to the bHLH ORANGE family, can rhythmically bind E-boxes after formation of CLK-CYC-PER complexes, thereby terminating CLK-CYC-mediated transcription of target genes including *cwo* itself[30–32]. The products of other

circadian genes, including *cryptochrome* (*Cry*), *vrille* (*Vri*), *Pdpε1*, and *double-time*, assist the CLK-CYC feedback loops and act synergistically to render and stabilize a circadian oscillation in diurnal or nocturnal rhythms[33–35]. Some circadian clock genes, such as *Pdpε1*, were found in *Drosophila* to be involved in the detoxification response, which has also been implicated in mammalian *Reb-Erbα*[36–38]. Whether the circadian network mechanism is shared in effecting downstream larval behavioral rhythms or insect adaptation to various host plants and related circumstances remains unknown.

In this study, we confirmed that *S. litura* larvae exhibited robust circadian rhythms in excretion and digestion from the 5th instar through the last (6th) instar, in synchrony with their typical rhythmic behavior of hiding in the soil during the daytime, coming out from the soil in the evening to eat crops throughout the night, and then returning to the soil with sunrise. Genome-wide RNA-seq analysis revealed distinctive circadian expression of genes for enzymes involved in digestion and detoxification in the midgut and fat body, as well as for 9 circadian rhythm genes newly annotated in this species. Functional studies including siRNA knockdown of selected circadian genes in vivo and overexpression in cultured cells provided evidence for their interaction with downstream detoxification genes via widely distributed promoter-element E-boxes. Altogether these studies provide a detailed look at mechanisms underlying circadian detoxification gene expression in a lepidopteran species and suggest an approach for insecticide treatment based on chronotoxicological considerations.

## Results

**Daily rhythms in locomotion and feeding of *S. litura* larvae.** *S. litura* last instar larvae exhibited a daily rhythm in locomotion and feeding during days 1–3 under LD (12 h light: 12 h dark) with the acrophase at midnight, ZT 18, and the trough at ZT 6 (Fig. 1a, b, Supplementary Data 4). The free-running rhythm of feeding under DD (continued darkness) was somewhat less pronounced but weak peaks continued with the acrophase being slightly earlier, perhaps due to $\tau$ being shorter than 24 h (Fig. 1b). Defecation activity was also rhythmic (Fig. 1c, Supplementary Data 4), peaking at ZT 9 during daytime in a 24 h cycle, suggesting that efficient digestion and excretion from the gut continued to occur in the day when the larvae appeared to be inactive or "sleeping". However, although the free-running rhythm in feeding exhibited clear peaks, the intensity was weaker. *S. litura* larvae showed a clearer rhythmic pattern in locomotor activity with a more conspicuous free-running rhythm than in feeding and defecation (Fig. 1a). The free-running rhythm in locomotion (Fig. 1a) seemed phase-delayed relative to the entrained rhythm, suggesting a circadian period, $\tau$, longer than 24 h, while that in feeding (Fig. 1b) was phase advanced, suggesting a $\tau$ shorter than 24 h.

**Rhythm of digestive enzymes in midgut of *S. litura*.** What follows feeding is sleep, digestion, and metabolism. Our question is whether the latter two processes are under circadian control. For 72 h of LD, we examined changes in the activity of α-amylase, total protease, and lipoprotein lipase in the intestinal fluid and found digestive enzyme activity in the midgut lumen oscillated (Supplementary Fig. 1). Similar to *S. littoralis*[15], the activity of α-amylase peaked at dawn. However, the total protease and lipoprotein lipase activity peaked at dusk, resembling the defecation rhythm. Furthermore, annotation of 6 digestion-related genes involved in carbohydrate (*Trehalase* and *Trehalose transporter*), protein (*Trypsin* and *Dipeptidase*), and lipid (*Lipase* and *Fatty acid binding protein*) metabolism in the midgut and measurement

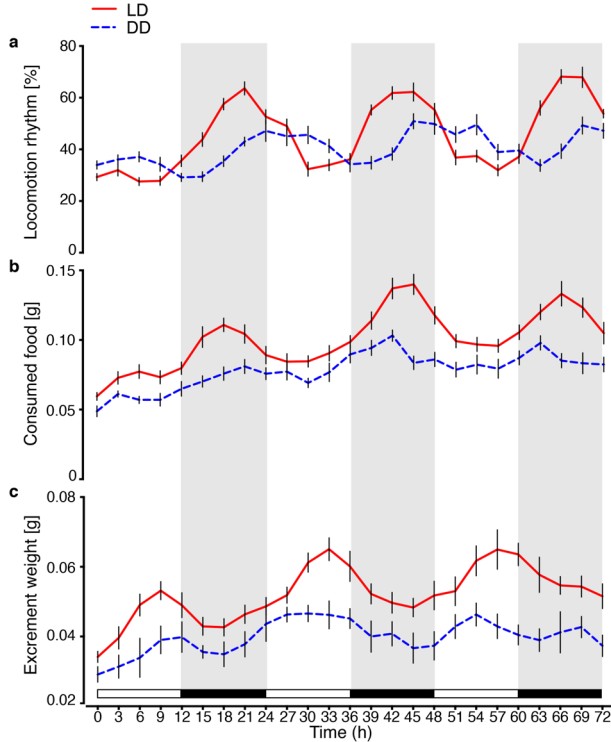

**Fig. 1 Behavioral rhythms of *S. litura* larvae during 6LD1 to 6LD3. a** Locomotion rhythm measured by recording the percentage of active larvae; **b** amount of consumed food; and **c** excrement weight under LD 12:12 (red solid line) and DD (blue dashed line) conditions. Nine groups each with 10 individuals were used for feeding behavior statistics. Counts were made for 6LD1-6LD3 larvae at 3 h intervals. Locomotion activity rhythm was measured according to the percentage of "active" larvae (see Methods); larvae were observed every 20 min, for a total of 9 values obtained at 3 h intervals. White and black shading indicates the photophase and scotophase. Error bars represent SEM.

of their transcriptional levels indicated higher expression during daytime (Supplementary Fig. 2). These data indicated that the life of this noctuid is characterized not only by nocturnal feeding but also intense activities of digestion during the day.

**Rhythmic expression of detoxification genes in the fat body and midgut.** Detoxification of abundant plant secondary compounds, much of which takes place in the fat body, is a vital process for survival on host plants. Detoxification of xenobiotics is also crucial for ecological adaptation to different host plants in highly polyphagous pests[7,39,40]. In *S. litura* and *S. frugiperda*, expansion of detoxification genes, such as cytochrome P450 (P450), carboxylesterase (COE), and glutathione *S*-transferase (GST), is reported to have enhanced the insects' detoxification ability[7,40]. Using RNA-seq, we performed a transcriptome analysis of 313 previously annotated detoxification genes in the midgut and fat body of *S. litura* 6th-instar larvae every 3 h during 24 h. These consisted of 138 *SlituP450s*, 47 *SlituGSTs*, 110 *Slitu-COEs*, and 18 *S. litura Aminopeptidase N* (*SlituAPNs*) based on the published *S. litura* genome[7]. Two hundred eleven detoxification genes were expressed in the midgut, and 220 genes in the fat body. Most of these genes oscillated during the 24 h sampling period. To look for differentially expressed patterns in the data, we performed clustering analysis using Mfuzz software with FCM parameters of $c = 12$, $m = 1.25$. This classified all expression profiles into 12 different clusters and yielded two main detoxification clusters in midgut and fat body, Cluster A11 and Cluster

B12 (Fig. 2a, Supplementary Fig. 3, Supplementary Data 4). We then plotted the peak expression profile of each detoxification gene in these two clusters against the time when the peak appeared (Fig. 2b, Supplementary Data 4). These patterns indicated that more than half (64.5%, Fig. 2b) of these detoxification genes were more highly expressed in daytime, i.e., ZT 3–6 (10 a.m.–1 p.m.) and ZT 9–12 (4 p.m.–7 p.m.) than at night with an antiphase to the feeding time. This suggested that the larvae could efficiently cope with the xenobiotic challenges in ingested food and insecticides during their inactive or "sleeping" daytime period.

In order to verify the reliability of the transcriptome data, we used RT-qPCR to determine the expression trend of a representative detoxification gene from each of the three analyzed families. These genes were selected because they were all highly induced by the neonicotinoid insecticide imidacloprid, as we reported previously[7]. Furthermore, their RT-qPCR patterns were consistent with the RNA-seq data (Fig. 2c, d, Supplementary Data 4).

**Circadian gene annotation and expression analysis.** To study the relationship of detoxification gene transcriptional activation with the circadian clock, we first annotated 9 core circadian genes, *SlituClk*, *SlituBmal1*, *SlituPer*, *SlituTim*, *SlituCwo*, *SlituCry1*, *SlituCry2*, *SlituVri*, and *SlituPdpε1*, in the *S. litura* genome by BLAST search with orthologous genes which have been reported in other lepidopteran insects (Supplementary Table 1). To investigate whether these identified clock genes were involved in the rhythmic output of feeding and locomotion, we then assayed their daily expression profiles using RNA-seq and RT-qPCR data measured systematically in three tissues/organs: head, midgut, and fat body (Supplementary Fig. 4). The results indicated that 6 of the annotated genes (*SlituPer*, *SlituTim*, *SlituCwo*, *SlituCry1*, *SlituCry2*, and *SlituVri*) exhibited 24-hour circadian rhythms in the head, while the peak time lagged in the midgut and fat body. Additionally, the profiles of *SlituClk* and *SlituBmal1* (and perhaps *SlituPdpε1*) showed a biphasic pattern in the head under LD.

To provide a more convincing demonstration of a circadian fluctuation in the expression of *SlituClk*, *SlituBmal1*, *SlituPer*, *SlituTim*, and *SlituCwo*, we extended the time for continuously checking the transcript levels in the brain under LD and DD conditions to 72 h at 3 h intervals (Fig. 3, Supplementary Data 1, 2). The expression of *SlituPer* and *SlituTim* manifested clear and strong rhythms in which peaks occurred at the middle of the dark phase, whereas, the *SlituCwo* rhythm was similar but slightly phase advanced relative to these two rhythms. *SlituCwo* showed a strong rhythm both in LD and DD; however, under DD, it was obvious that *SlituTim* did not free run, nor probably did *SlituPer*. In contrast, *SlituClk* and *SlituBmal1* showed a 12-hour rhythm with a peak in each 12 h segment and weaker intensity under free running conditions. In addition, *SlituActin3*, used as a reference gene in RT-qPCR, maintained a consistently level expression with no change throughout day or night (Fig. 3).

**Effects of siRNA injection on the expression of detoxification genes.** In order to determine the relationship between clock genes and their potential downstream clock-controlled genes we performed knockdown experiments for *SlituClk*, *SlituBmal1*, and *SlituCwo*, singly or collectively, by injecting 6th instar larvae with siRNA. Knockdown with combined siRNAs for *SlituClk* and *SlituBmal1* caused a 36–42% decrease of transcription levels for representative detoxification genes *SlituP450-082*, *SlituP450-132*, *SlituGST022*, *SlituGST035*, *SlituCOE051*, and *SlituCOE092* compared to controls injected with siRNA for green fluorescent protein (GFP) (Fig. 4a, Supplementary Data 4). On the contrary,

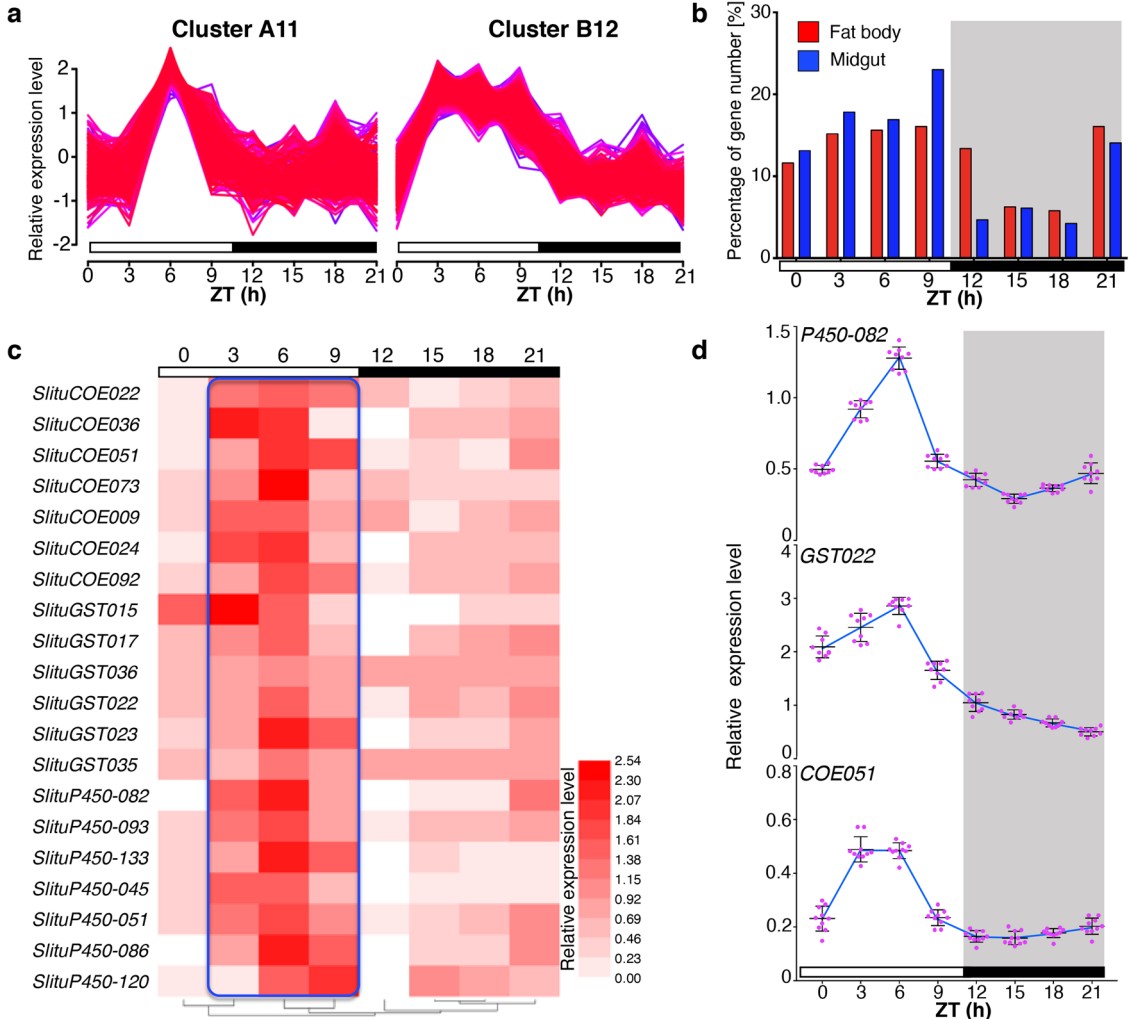

**Fig. 2 Oscillation of detoxification gene expression in the fat body and midgut. a** Clustering of genes with the same expression pattern in the midgut (Cluster A11) and fat body (Cluster B12). The expression values were processed using Mfuzz software with the method of homogenization by log10. **b** The percentage of detoxification genes with peak expression at specific time points in the fat body and midgut. **c** Expression heatmap of selected detoxification genes. **d** RT-qPCR analysis of representative detoxification genes from each of the three major families, *P450-082* (top), *GST022* (middle), and *COE051* (bottom). Each experiment was performed with fat bodies from 3 6LD2 larvae and was repeated independently three times. Error bars represent SEM, photophase is represented by white rectangles, and scotophase by black rectangles and black shading.

knockdown of *SlituCwo* alone enhanced expression of these detoxification genes by 37–46% compared to controls (Fig. 4b, Supplementary Data 4).

To test whether the observed alteration of putative detoxification gene expression affected insecticide sensitivity, at 24 h after siRNA injection we fed larvae artificial diet containing imidacloprid ($30\,\mu g\,g^{-1}$), followed by measuring the percent of affected larvae in each group at 6, 12, 18, and 24 h after feeding the insecticide. We observed an increase (maximum of 69.6%) in sensitivity of the *Clk* + *Bmal1* knockdown group compared to the control. By contrast, SlituCWO acted as a repressor whereby its knockdown decreased (around 36.1%) the sensitivity to the insecticide (Fig. 4c, d, Supplementary Data 4).

**E-box involvement in the circadian transcriptional activation of detoxification genes.** While examining transcriptional sequences and promoter regions in the genome data, we identified abundant conserved putative *cis*-regulatory elements on the promoters of some detoxification genes, including the six representative ones tested in knockdown experiments. Therefore, as potential binding sites for transcription factors we further

annotated the locations of E-boxes (CACGTG) and non-canonical E-boxes (CANNTG) in the 5′ regulatory regions of all expressed detoxification genes, which included 142 daytime activated detoxification genes, 57 night activated detoxification genes, and 21 detoxification genes with no fluctuation (Supplementary Data 3, Fig. 5). It was evident that arrhythmic detoxification genes, without fluctuating expression, had few E-boxes (0.57 E-box per gene), clearly suggesting E-boxes are associated with rhythmic gene expression. Accordingly, E-box elements were more abundant (2.54 E-box per gene vs 1.77 E-box per gene) in the promoter regions of detoxification genes that were activated during the day compared with those activated at night. This was especially evident for some subfamilies previously reported to have efficient detoxification function and active response to toxins such as P450 Clan3, GST class ε, and lepidopteran type COE[7] (Supplementary Data 3). In addition, clustered detoxification genes which were activated and highly expressed at the same time during the day had more E-box elements in their promoter regions within 3 kb upstream of their transcriptional start sites (Fig. 5a, Supplementary Data 3), which may be the key reason for their simultaneous activation and expression during the day.

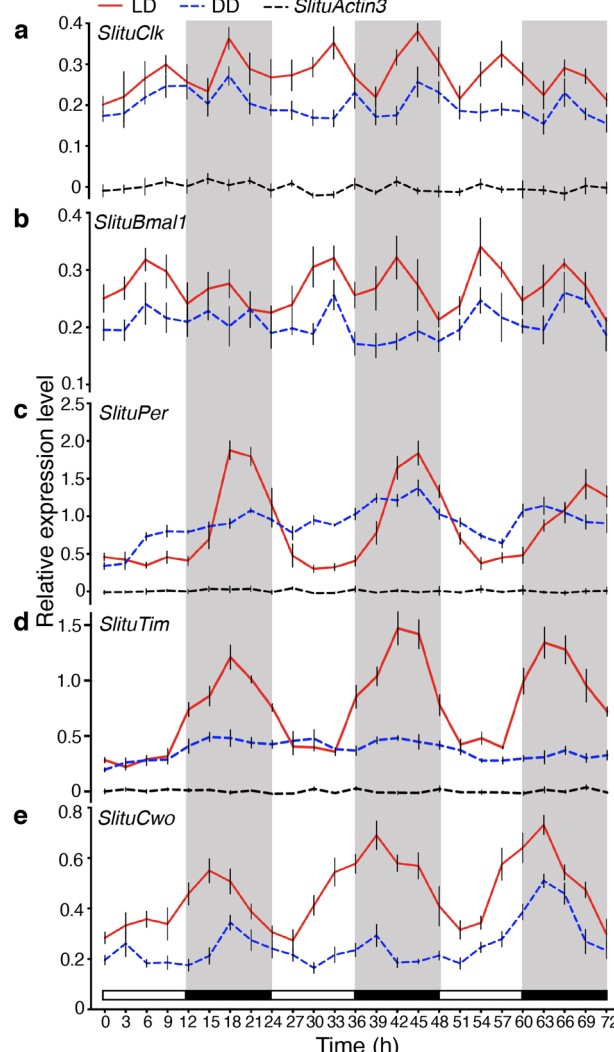

**Fig. 3 RT-qPCR analyses of core circadian gene transcription in larval brain.** Amounts of transcripts of *SlituClk* (**a**), *SlituBmal1* (**b**), *SlituPer* (**c**), *SlituTim* (**d**), and *SlituCwo* (**e**) during 6LD1 to 6LD3 under LD 12:12 (red solid line) and DD (blue dashed line) after entrainment by LD 12:12. Brains were collected from 3 groups with 10 individuals each at 3 h intervals for 72 h under the two conditions. The expression profile of the reference gene *SlituActin3* (black dotted line) is shown after homogenization with log10. Light and black shade represent the photophase and scotophase. The results are given as mean ± SEM of three independently repeated experiments.

Moreover, E-boxes associated with detoxification genes that were most efficiently transcribed in the daytime were mainly concentrated in the upstream 300–1800 bp (Fig. 5b, Supplementary Data 3).

To confirm whether circadian factors could bind directly to these enhancer elements on detoxification gene promoter regions, we performed a co-transfection assay in the *S. litura* embryonic cell line Spli-221 using overexpression vectors containing individual sequences of *SlituClk*, *SlituBmal1*, *SlituPer*, and *SlituCwo*, together with dual-luciferase reporter vectors that contained E-box sequences in the promoters of putative target genes representing three detoxification gene families, *SlituP450-082*, *SlituGST035*, and *SlituCOE051* (Fig. 6a, b, Supplementary Fig. 5, Supplementary Data 4). As expected, the overexpression of circadian factors SlituCLK and SlituBMAL1 increased transcription of these representative detoxification genes tested by an

average of 58.4%. However, the promoter activities of the test detoxification genes decreased by an average of 34.2% after transfection with the *SlituPer* overexpression vector. This finding was consistent with our previous observations of the likely role of SlituPER as a repressor and indicated that accumulated SlituPER protein could inhibit E-box binding of a putative CLK-BMAL1 heterodimer to mediate its own transcriptional repression and that of other E-box-dependent genes. By contrast, in the cells containing overexpressed *SlituCwo*, we detected a much lower level (average reduction of 124.1%) of test gene transcription. This suggested that SlituCWO might also act as a typical repressor involved in the regulation of the detoxification rhythm.

To test further whether circadian protein binding to E-box elements could alter the transcription of detoxification genes, we constructed a mutant vector for the promoter region of *SlituP450-082*, replacing the 3 normal E-box elements (CACGTG) with TGTACA. The transcriptional activity of the mutant was lower (62.4%) than that of the positive control (Fig. 6c, Supplementary Data 4). This indicated that the presence of intact E-boxes was crucial for the efficient expression of detoxification genes, consistent with their proposed function as target sites for circadian rhythm regulators to control the periodic expression of detoxification genes.

**Chronotoxicological consideration in agriculture: detoxification activity and sensitivity to pesticides**. To further confirm the existence of a rhythmic xenobiotic detoxification process and determine optimal conditions for future insecticide applications in the field, we treated two groups of fourth-instar larvae topically by directly placing a solution of imidacloprid on their dorsal surface once each at a day or night circadian time under LD and DD conditions, followed by observations of the response rate (including mortality) over a 6 h period. The sensitivity to the same dose of insecticide averaged 21.6% higher when applied nocturnally than diurnally, indicating night to be a more effective time for use of agricultural pesticides (Fig. 7, Supplementary Data 4).

## Discussion
Genetic systems supporting the core circadian system are basically conserved from Archaea to human[13,26,29]. These are composed of an interlocked negative feedback system consisting of two loops. The first, core feedback loop is composed of two arms: a positive arm comprising CLK:BMAL1 in mammals and monarch butterfly and CLK:CYC in *Drosophila* which activates transcription of target genes, and a negative arm in which the negative elements PER/CRY in mammals and monarch, and PER/TIM in *Drosophila*, repress their own CLK:BMAL1-mediated transcription and those of other target genes. The second, stabilizing loop relies on *RevErbα/Rorα* in mammals and *Pdpε1/Vri* in insects[10,20,30,33,34,41–43]. Feeding and light cycle are two major cues that drive the circadian system, but feeding is accompanied by a massive flow of secondary metabolites and xenobiotic agents[44]. All CYPs require heme as a prosthetic group and heme availability is strongly circadian[19,38], consistent with strong circadian control of their activity. Redox metabolites must depend on the energy flow starting with feeding, and effective feeding is guaranteed by the circadian system[19,38]. Thus, the coupling of xenobiotic responses and digestive mobilization must be adaptive.

In this report we introduce a molecular model for the coupling of behavior and metabolism in a formidable lepidopteran pest. Genome-wide transcriptomic and functional analyses in *S. litura* revealed a complementarity between periods of intensive digestion and detoxification activity during the daytime and

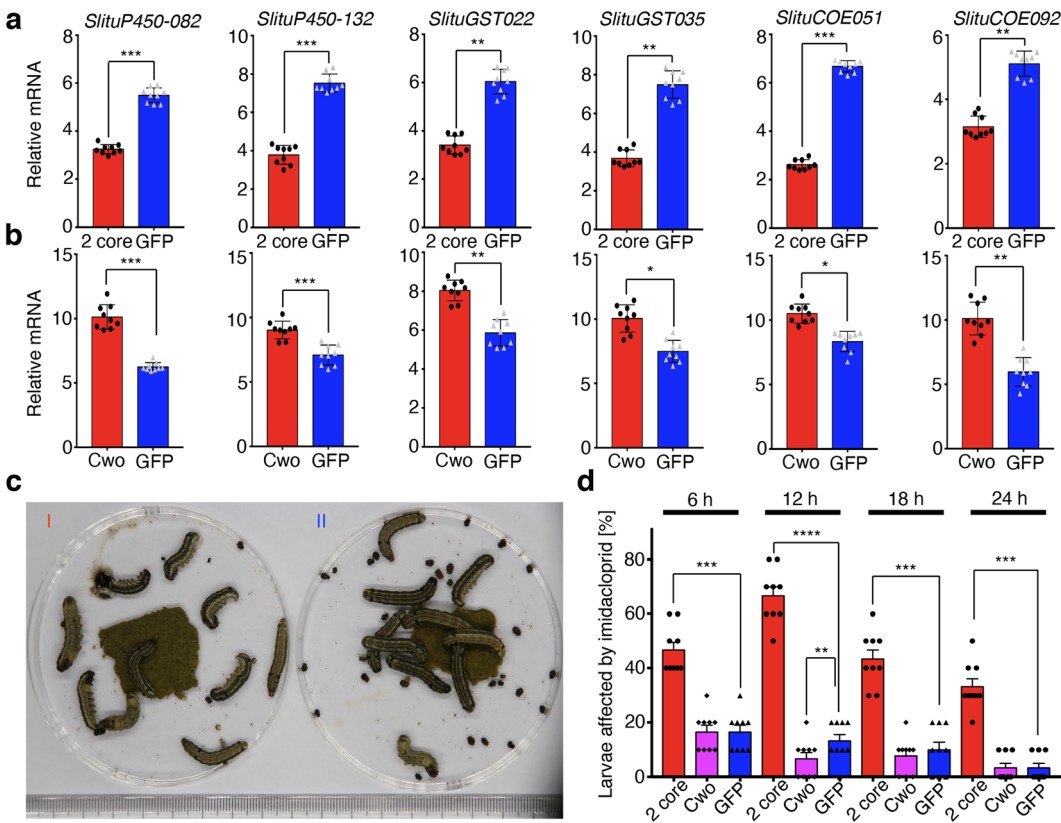

**Fig. 4 Effects of knockdown of circadian core genes on expression of detoxification genes and imidacloprid sensitivity.** Relative expression of detoxification genes was measured by RT-qPCR after injection with **a** combined siRNAs for *SlituClk* and *SlituBmal1* ("2 core", red) or siRNA for *GFP* (control, blue); and **b** siRNA for *SlituCwo* (red) or *GFP* (control, blue). **c** Effect of artificial diet containing imidacloprid (30 μg/g) on larvae after the injection of siRNA with [I] combined *SlituClk + Bmal1* or [II] *GFP* as control. **d** Time course of imidacloprid sensitivity after knockdown. Larvae were fed artificial diet supplemented with imidacloprid (30 μg/g) 24 h after knockdown by siRNA injection; "affected" larvae were recorded every 6 h. Larvae were scored as "affected" when they rounded up and did not move or feed and excreted shapeless feces. Several hours later, some recovered from this "suspended" state and some died. Three groups of 10 6LD1 individuals were prepared for siRNA injection and each value represents the average of three experiments with SEM error bars. The level of statistically significant difference was set at *P value < 0.05, **P value < 0.01, ***P value < 0.001 and ****P value < 0.0001.

locomotion and feeding at night under control of a circadian system. Rhythmic feeding behavior is associated with the rhythmic activity of digestion enzymes accompanied by rhythmic expression of circadian genes in a previous report on *S. littoralis*[15]. Although these and other polyphagous noctuid pests like *S. frugiperda*, *H. armigera*, and *H. zea* contain a great expansion of gene families associated with detoxification of plant secondary metabolites and insecticides[1,2,7], so far few studies have documented the transcriptional activity of these detoxification genes in relation to their distinctive circadian behaviors, nor have they elucidated the molecular mechanisms underlying these processes in Lepidoptera.

The transcription patterns of most *S. litura* orthologs of *Drosophila* circadian genes were rhythmic, at least under an LD cycle. However, the patterns of *SlituClk* and *SlituBmal1* were biphasic (Fig. 3a,b), differing from *DmClk* which has an antiphase pattern to *DmPer* transcription[31]. *SlituPer* and *SlituTim* had the same acrophase (Fig. 3c, d) and therefore are likely partners as in *Drosophila* where *DmPer* and *DmTim* mRNA levels start to rise at CT 4 during the natural daytime and reach peaks within 3 h of dusk[25,31]. Although *SlituPer* and *SlituTim* showed rhythmic expression with an acrophase at ZT 18 (Fig. 3c, d), their peak expression was delayed compared to *Drosophila*, and increased gradually from midday to midnight.

The entrained rhythms of *SlituPer* and *SlituTim* had high amplitudes, but they did not free run in DD. This failure suggests the possibility that they are photo-induced and function together

as an hourglass timer, so that alternative devices exist to measure time endogenously as changeable settings[11]. By contrast, *SlituCwo* transcription showed a robust rhythm with high amplitude under an LD cycle and was partially rhythmic in the DD condition (Fig. 3e). *SlituPer* mRNA and *SlituCwo* mRNA were not in phase in the acrophase, suggesting that *SlituCwo* may play a role similar to that reported in *Drosophila*[45], in which CWO rhythmically binds E-boxes to promote PER-dependent removal of CLK-CYC and maintain repression of transcription. For now, without more evidence for the detailed network within the feedback loops of *S. litura*, it is difficult to interpret the effects of an individual knockdown targeting one member of the circadian system. Nevertheless, the results reported here, together with the results of the mixed siRNA injection targeting *SlituClk* and *SlituBmal1* and in vitro E-box modification studies, confirm the existence of a molecular link between the circadian system and detoxification rhythm in *S. litura*.

The reduced transcription of representative detoxification genes by larval injection of combined siRNAs targeting core circadian genes *SlituClk* and *SlituBmal1* (Fig. 4) suggested their diurnal expression is most likely driven by a circadian system. These observations were supported by the finding of multiple E-boxes on members of 3 families of detoxification genes along with experiments which showed that overexpression of *SlituClk* and *SlituBmal1* upregulated detoxification gene transcription in cultured cells, while SlituCWO acted as a repressor to downregulate their transcription (Fig. 6). Thus, we propose that the

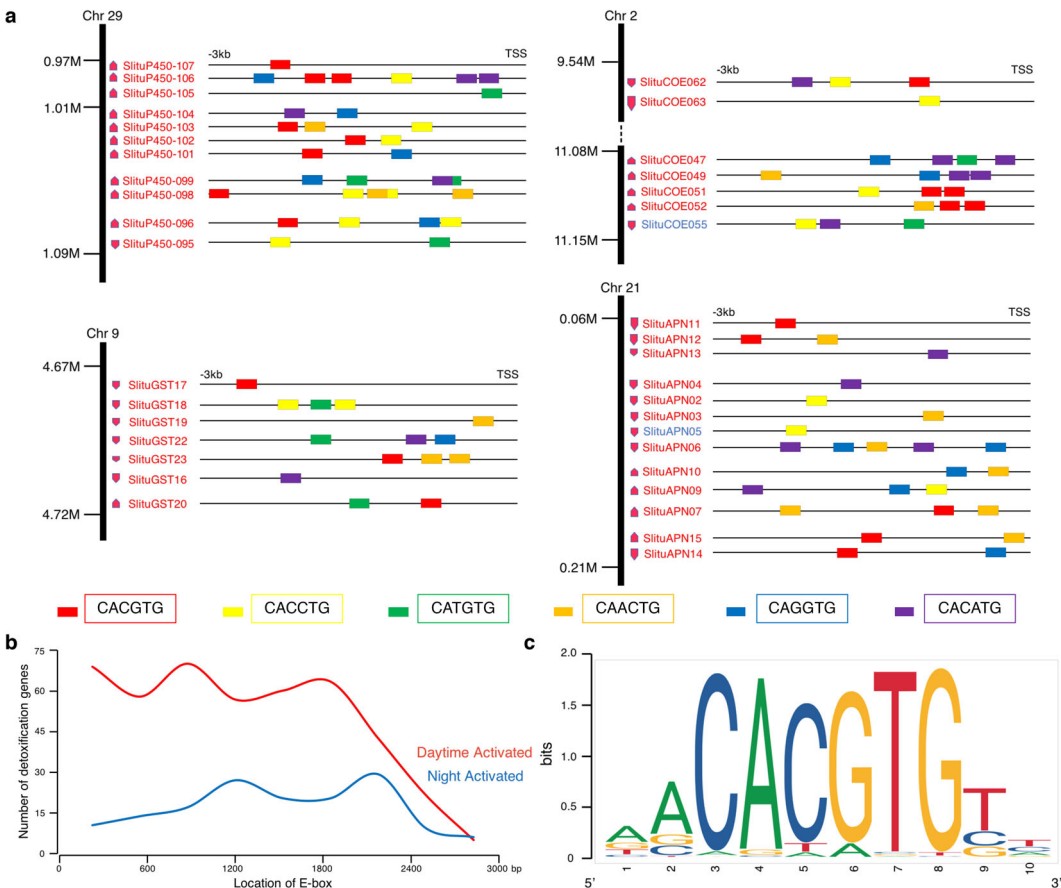

**Fig. 5 E-box annotation and location in the regulatory region of detoxification genes. a** Location of multiple E-boxes in the regulatory regions of clustered detoxification genes. Different types of E-boxes are annotated 3 kb upstream from transcriptional start sites (TSS) and shown with different colored rectangles. Daytime activated detoxification genes (red) and night activated detoxification genes (blue) are shown for a given cluster. **b** The number of detoxification genes relative to the distance between the E-box sequence and the TSS. **c** Annotated motifs of canonical E-boxes recognized by specific transcription factors.

heterodimer, SlituCLK-SlituBMAL1, as well as SlituCWO, can activate or repress transcription of detoxification genes competitively by binding to E-boxes to activate or mediate output behaviors under regulation of the circadian system. Given the knockdown results (Fig. 4), we propose that core genes *SlituClk* and *SlituBmal1* are associated with *SlituCwo* as a negative regulator to control the oscillation of the detoxification process. E-boxes are especially enriched in the regulatory region of clustered detoxification genes, such as *SlituP450-095~SlituP450-107* located on Chr29 (Fig. 5). In contrast to the night activated detoxification genes, which are scattered in the genome, daytime activated detoxification genes are neighboring and close enough in their clusters to be within the same chromatin loop[7]. This structure could contribute to synchronously and efficiently initiating transcription of clustered detoxification genes via chromatin domain activation through binding of clock factors to E-box sequences and mediating the access of RNA polymerase to their transcriptional start sites[46].

Multiple E-boxes or non-canonical E-boxes were also found in the 5′ regulatory regions of the six digestion enzyme genes previously documented to show diurnal fluctuation in expression (Supplementary Fig. 6a). Although we did not test the circadian regulation of their transcriptional activity directly, the finding that altering the canonical E-box sequences of genes for digestion of carbohydrate and lipid was accompanied by abolishment of their activity in cultured cells is consistent with their involvement with circadian mechanisms similar to those documented for

detoxification genes (Supplementary Fig. 6b, Fig. 6), and merits further study.

Biological clocks are reported to control various behaviors and physiological processes in Lepidoptera such as locomotor activity, feeding rhythms, mating behaviors, and other developmental and metabolic events[10,11,15,16,47]. In this study, we not only analyzed the daily rhythm of larval activities and feeding behavior, but also strongly focused on detoxification metabolism, which was closely related to feeding. Our investigation showed that *S. litura* displayed circadian rhythms in the expression of detoxification genes as well as in insecticide sensitivity using the neonicotinoid imidacloprid. Similar findings have been reported in other insects. In fruit flies, the maximum expression of many xenobiotic metabolizing genes clustered in late afternoon, while the daily profiles of susceptibility to pesticide indicated increased resistance in midday in *D. melanogaster*[17]. In a study of *Anopheles gambiae*, the GST activity of Pimperena strain mosquito lysates had peak phases at late-night to dawn, whereas in insecticide assays using the Mali strain, the mosquitoes showed rhythmic susceptibility to DDT with a peak at late afternoon[18]. In exploring the daily rhythms of insecticide susceptibility in the bedbug, *Cimex lectularius*, Khalid et al. reported that the nocturnally active larvae repeatedly showed highest tolerance for deltamethrin during the late photophase at ZT9[48]. Other insects such as spotted wing *Drosophila*, cotton aphid, and brown plant hopper[49–51] are also reported to show circadian fluctuations in their detoxification metabolism and insecticide susceptibility. But as yet none of these

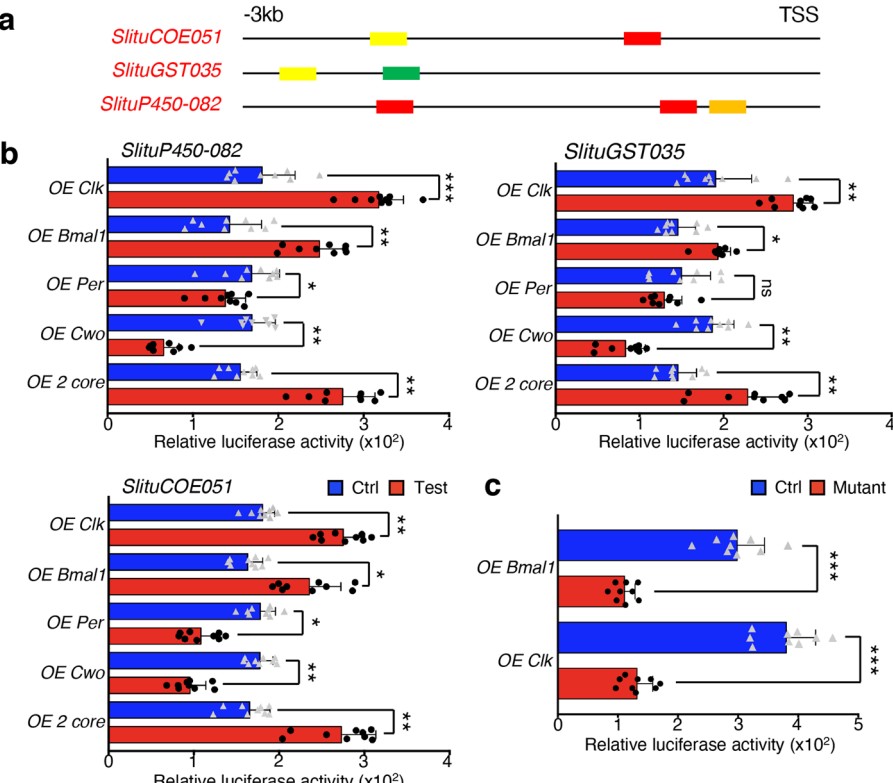

**Fig. 6 Effects of core circadian gene overexpression on detoxification gene transcription through E-box binding in cultured cells. a** Location of canonical (CACGTG, red) and non-canonical (CACCTG, yellow; CATGTG, green; CAACTG, orange) E-boxes in 5′ regulatory regions of *SlituCOE051*, *SlituGST035*, and *SlituP450-082*. **b** Relative luciferase activity for the promoters of *SlituP450-082*, *SlituGST035*, and *SlituCOE051* induced by co-transfection of Spli-221 cells with *SlituClk* (*OE Clk*), *SlituBmal1* (*OE Bmal1*), *SlituPer* (*OE Per*), *SlituCwo* (*OE Cwo*), and combined *SlituClk* + *Bmal1* (*OE 2 core*) overexpression vectors (red). Control vectors overexpress *EGFP* (blue). **c** Relative luciferase activity of mutated (TGTACT, red) and normal (CACGTG, Ctrl, blue) E-box sequences for *SlituP450-082* induced by co-transfection of *SlituClk* and *SlituBmal1*. The results are given as mean ± SEM of three repeated experiments and statistically significant difference was set at *$P$ value < 0.05, **$P$ value < 0.01 and ***$P$ value < 0.001.

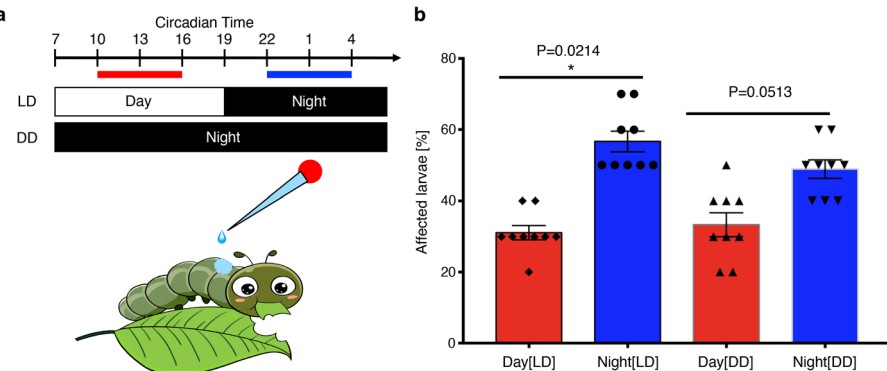

**Fig. 7 Topical treatment of imidacloprid on 4th instar larvae. a** Outline of day–night settings and timing of imidacloprid treatment. **b** The effect of imidacloprid exposure on normal larvae. The experiments were performed during daytime (10 a.m.–4 p.m., red) and night (10 p.m.–4 a.m., blue) in normal LD (12 h:12 h) and DD (continued darkness) conditions based on a preliminary $LD_{50}$ test (see Methods). Larvae were scored as "affected" when they rounded up, stiffened and did not move when touched, as if dead ("suspended animation"). Three groups of 10 4LD2 larvae were used; each group was treated once during the day or night. The mean percentage of larvae affected after direct exposure to imidacloprid solution on their dorsal surface is shown ± SEM with three repeated tests and statistically significant difference was set at *$P$ value < 0.05, **$P$ value < 0.01 and ***$P$ value < 0.001.

studies have reported details of mechanisms driving these processes.

Our study started with the rhythms of behaviors and then found a duality between rhythms of detoxification metabolism and feeding behavior. Further experiments showed that this duality was under the regulation of the circadian system, through binding of circadian elements to canonical E-boxes on responding downstream genes. We propose a molecular link between circadian clocks and daytime xenobiotic detoxification contributes to intensive larval night feeding in *S. litura* which enables it to evade the risks of insecticides and possible predators in daytime. Given the geographically widespread agricultural damage of this formidable insect, our results advance our understanding of the molecular basis underlying its ability to adapt to diverse adverse

environmental conditions and promise to contribute to more effective management of this pest.

## Methods

**Larvae preparation and sample collection.** Larvae were from the inbred Ishihara strain described in a previous paper[7]. All larvae were reared on artificial diet (Silk Mate, Japan) at $25 \pm 2\,°C$ with $60 \pm 5\%$ relative humidity under 12 h: 12 h light-darkness (LD) conditions where Zeitgeber time 0 (ZT0) is the time of lights on and ZT12 is the time of lights off. Since larvae began to show circadian rhythms in feeding behavior from the 5[th] instar, 6LD1 (6th instar day1) to 6LD3 larvae were used for all experiments. For transcriptome analysis, heads, fat bodies, and midguts from 6LD1 larvae were dissected at 3 h intervals for 24 h. Tissues were rinsed once in PBS buffer, transferred to Trizol reagent (Invitrogen, USA), and stored at $-80\,°C$ until use. Each sample had 3 replicates for a total of 9 individuals at each time-point.

**RNA extraction and sequencing.** Total RNA of the head, midgut, and fat body was extracted from excised tissues using Trizol reagent according to the manufacturer's instructions (Invitrogen, USA), and contaminating genomic DNA was digested with RNase-free DNase I (Takara, Japan). Approximately 1 μg total RNA was used to construct cDNA libraries using a TruSeq RNA sample preparation kit (Illumina, USA). The library was sequenced with an Illumina HiSeq4000 system (Illumina, USA).

**Transcriptome analysis.** After removal of polyA, rRNA, tRNA, and low-quality reads (QV < 20) from raw reads as described in a previous paper[52,53], RSEM software (v1.2.12) was used to count the number of mapped reads and estimate the FPKM (fragments per kilobase per million mapped fragments) value of each gene as a measure of the gene expression level. Significant differential expression of genes was determined using a false discovery rate of <0.01 and a ratio of intensity against control of >2 for induction or <0.5 for reduction. We integrated specifically differentially expressed genes (DEGs) into an expression matrix displayed as heatmaps using R (Logiciel) (https://www.r-project.org/). We performed clustering analysis using Mfuzz software with FCM parameters of $c = 12$, $m = 1.25$ (http://mfuzz.sysbiolab.eu), which is a package based on R with a soft clustering algorithm.

**Monitoring feeding behavior and locomotor rhythm.** On 6LD1, nine groups each with 10 individuals were prepared for feeding behavior statistics. Each group was fed 10 g of artificial diet in LD or DD conditions. Locomotion activity was recorded by a video monitoring system which consisted of a video recorder, an automatic light switch, a networked infrared camera, and a display screen. The movement distance was estimated as the number of times a larva crossed grid lines 2 cm apart on a $20 \times 20$ cm plate at 20 min intervals for 24 h or more. Based on a pre-experiment, the standard for judging "activity" was that a larva crossed the grid lines more than 12 times in 20 min. The amount of diet consumed and excrement weight were measured at 3 h intervals from 6LD1 to 6LD3 and fresh diet was introduced after taking each sample. All data of feeding, excrement weight, and movement are deposited in Supplementary Data 4.

**Measurement of digestive enzyme activity.** Midguts containing fluid were dissected at 3 h intervals from 6LD1–6LD3 larvae and frozen in liquid nitrogen, then stored at $-80\,°C$ until further use for measurement of α-amylase activity, total protease activity, and lipoprotein lipase activity. The 3,5-dinitrosalicylic acid (DNS) method was used for α-amylase activity which involved mixing 500 μl activity buffer with 10 μl sample supernatant with the addition of 1% starch as substrate for 1 h incubation at $37\,°C$[54]. The reaction was stopped by adding DNS reagent and heating in a metal bath for 10 min at $100\,°C$. The supernatant α-amylase activity was quantified spectrophotometrically at 540 nm and calculated as the amount of reducing sugars from starch digestion in 1 h. For total protease activity, 10 μl midgut samples were incubated with 2% azocasein as the substrate in activity buffer for 1 h at $37\,°C$[55]. The reaction was stopped with 30 μl of 12% trichloroacetic acid and allowed to precipitate at $4\,°C$ for 2 h. After adding an equal volume of 0.5 M NaOH to adjust the color of the supernatants, the protease activity was quantified spectrophotometrically at 595 nm. One unit (U) proteolytic activity was defined as the amount of enzyme required to increase the $OD_{595}$ by 0.1 per min in 1 ml of mixture. Lipoprotein lipase activity was measured with a lipoprotein lipase assay kit (abcam, UK) in accordance with the manufacturer's protocol. One unit (U) LPL activity was defined as the amount of lipoprotein lipase that generated 1.0 nmol of fatty acid product per min at pH 7.4 and $37\,°C$.

**Gene annotation.** Nine core circadian clock genes including *Clock* (*Clk*), *brain and muscle arnt-like 1* (*Bmal1*), *Period* (*Per*), *Timeless* (*Tim*), *Clockwork orange* (*Cwo*), *Cryptochrome1* (*Cry1*), *Cryptochrome2* (*Cry2*), *Vrille* (*Vri*), and *PAR domain protein ε1* (*Pdpε1*) were manually annotated to identify intron/exon boundaries[7]. The gene ID and intron/exon boundaries are listed in Supplementary Table 1.

**Quantitative RT-PCR.** To quantify the temporal (24 h period) expression of core circadian clock and other candidate genes, we conducted RT-qPCR experiments

with the mRNA from heads, fat bodies, and midguts of *S. litura* 6LD1 larvae collected at 3 h intervals. The primers for target genes were designed using the software Primer 5.0 (Supplementary Data 1). After evaluating the quality and quantity of total RNA, first-strand cDNA was synthesized from 1 μg total RNA using a PrimeScript RT reagent kit with gDNA Eraser (Takara) according to the manufacturer's instructions. RT-qPCR was performed using a SYBR Green Supermix (Takara) in accordance with the manufacturer's instructions. *SlActin3* was used as a control for each set of RT-qPCR reactions as previously published[7] (Supplementary Data 1). The raw data for RT-qPCR results are presented in Supplementary Data 2.

**siRNA knockdown and insecticide response experiments.** Larvae (6LD1) were injected with siRNA for circadian clock and detoxification genes (as reported above) followed by measuring corresponding mRNA levels with RT-qPCR at 3 h intervals. Five μl of siRNA with a concentration of 100 pmol $μl^{-1}$ was injected into the hemolymph of each larva; injection of the same amount of *GFP* siRNA was used as control. siRNA injection was performed on groups of 10 individuals each for three replicates. siRNA sequences are listed in Supplementary Data 1. To determine the insecticidal effect of imidacloprid ingestion, larvae were fed an artificial diet (Silk Mate, Japan) supplemented with imidacloprid at 30 μg $g^{-1}$ 24 h after injection. The insecticide concentration was determined by preliminary tests as the approximate dose causing a visible effect (see below) without killing the larvae. Numbers of affected larvae and detoxification gene expression patterns were compared with the control groups at 3 h intervals during a 24 h period. "Affected" larvae rounded up, stiffened, and did not move when touched, as if dead (suspended animation); however, many affected larvae recovered from this suspended state after several hours likely due to detoxification of ingested imidacloprid.

**E-box annotation.** For clock-controlled candidate genes, we performed binding site analysis on sequences distal to the promoter regions flanking 3 kb of protein coding gene transcription start sites (TSS). Then each binding site that contained an E-box or tandem E-boxes was annotated with the Ensembl transcript having the closest TSS using the R package biomaRt (https://www.bioconductor.org/packages/release/bioc/html/biomaRt.html).

**Cell culture and dual-luciferase reporter assay.** The *S. litura* embryonic cell line Spli-221 was maintained in Grace medium (Thermo, USA) supplemented with 10% HyClone fetal bovine serum (GE Healthcare Life Sciences, USA) at $27\,°C$. The ORF sequences for *SlituClk*, *SlituBmal1*, *SlituPer*, and *SlituCwo* (see GenBank under BioProject PRJNA344815) were subsequently cloned into the *S. litura* embryonic cell line Spli-221 with the shuttle vector pSLfa1180fa modified and stored in our laboratory[56]. As a negative control a vector was also constructed for overexpressing *EGFP* in Spli221 cells. Promoter-specific primers were designed for cloning potential promoter regions that harbored one or tandem E-box elements located up to around 3 kb upstream of TSS sites (Supplementary Data 3). After restriction enzyme digestion and purification, the fragments were cloned into the plasmid pGL3-basic (Promega, USA) and used to construct luciferase reporter vectors. Transfection of the recombinant vectors was performed using the Cellfectin II reagent (Thermo, USA) according to the manufacturer's instructions. After an additional 48 h of incubation, cells were collected for RT-qPCR analysis of the target genes, and subjected to a luminometer-based dual-luciferase assay (Promega, USA) in accordance with the manufacturer's protocol.

**Western blotting.** To test whether the circadian proteins (SlituCLK, SlituBMAL1, SlituPER, and SlituCWO) were successfully overexpressed in Spli-221 cells, we performed Western blotting analysis after 48 h transfection. The cells were extracted in lysis buffer (300 mM NaCl, 3 mM $MgCl_2$, 100 mM Hepes-NaOH, 20% glycerol, 1 mM EDTA, 1% sodium deoxycholate, 2% TX-100, 0.2% SDS, pH 7.5) and the protein concentrations were estimated using a bicinchoninic acid (BCA) assay (Beyotime, China). Twenty μg protein per sample was separately resolved on 12% SDS-PAGE gels and subsequently transferred to PVDF membranes (GE Health Care), followed by immunoblotting using primary antibodies targeting HA (or FLAG) and anti-mouse IgG(H + L) antibodies (Sigma, USA). The signal was visualized with chemiluminescence using a SuperSignal West Femto Maximum Sensitivity Substrate kit (Thermo, USA).

**Topical treatment to test insecticide sensitivity of fourth-instar larvae.** The $LD_{50}$ of fourth-instar larvae was previously measured with concentrations at 0.5, 0.1, 0.05, 0.01, 0.005, and 0 mg/ml. Two μl of imidacloprid solution (at approximate $LD_{50}$ concentration of 0.01 mg/ml in acetone) was placed on the dorsal surface of each larva at the beginning of the day or night circadian cycle and the toxin sensitivity was recorded in two time frames, from 10 a.m. to 4 p.m. and from 10 p.m. to 4 a.m. under LD and DD conditions. After six hours, infection rate (including dead larvae) statistics were determined as *P* values described in the section *Statistics and reproducibility*. Three groups of 10 4LD1 larvae were used for $LD_{50}$ measurement and topical treatment and the experiments were repeated three times.

**Statistics and reproducibility**. Results are expressed as the means ± SEM. Statistically significant differences were evaluated using the Student's $t$ test for unpaired samples. The level of statistically significant difference was set at *$P$ value < 0.05, **$P$ value < 0.01, ***$P$ value < 0.001, and ****$P$ value < 0.0001. For analysis of rhythmic expression patterns, the nonparametric test, JTK_CYCLE, was used to incorporate a window of 12–24 h for the determination of circadian periodicity[57,58]. Appropriate sample sizes are reported under specific methods and in figure legends. Three groups with 10 individuals were used for knockdown experiments and topical treatment and 9 groups with 10 individuals were used for behavioral monitoring. RT-PCR analysis and co-transfection for E-box binding assay were performed with three independent replicates.

**Reporting summary**. Further information on research design is available in the Nature Research Reporting Summary linked to this article.

## Data availability

RNA-Seq data for transcriptome analysis have been submitted to the NCBI Short Read Archive under accession number PRJNA511360 and the annotated sequences have been submitted to GenBank under BioProject accession PRJNA344815. All other data generated or analyzed during this study are included in this published article and its Supplementary Information and Supplementary Data files. All relevant data are available from corresponding author upon request (to mitakazuei@gmail.com).

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

## Acknowledgements

This research was supported by a grant from the One Thousand Foreign Experts Recruitment Program of the Chinese Government (No. WO20125500074) and by the Foundation Project of Southwest University (SWU019033).

## Author contributions

J.Z., M.R.G., M.T., and K.M. conceived the project design and wrote the manuscript. J.Z. performed all the experiments. J.Z., S.L., and W.L. performed analysis of RNA-seq data. Z.C., H.G., and J.L. helped to perform RT-qPCR experiments. Y.X., Y.X., and L.Z. helped to perform knockdown experiments. K.P.A., G.S., and Q.X. helped to revise and comment on the manuscript.

## Competing interests

The authors declare no competing interests.
