## [Peer Review File · Communications Biology]

Reviewers' comments:

Reviewer #1 (Remarks to the Author):

Zhang et al. reports on a well-designed and executed set of inter-related experiments on the circadian system and regulation of physiology and behavior in *Spodoptera litura*. The manuscript is well written and easy to follow. The study characterizes and explores the relationship between the putative circadian clock genes and rhythmic detoxification processes. It also provides a translational component with consideration of time-specific application of pesticides. The work includes a semi-quantitative analysis of behavioral rhythms and of clock gene rhythms. The results and conclusions are compelling. Apart from a few minor corrections to the text, the paper is sound.

Minor comments

Line 108. Grammar. 'Strongly' does not work well in this sentence. Perhaps 'robust' would be a better choice?

Line 152. Insert 'a' between 'it' and 'most'.

Line 213. The authors have a misconception regarding the organization of the molecular clock. Clk and cyc form the positive loop, and per/tim the negative loop. As such this sentence should be corrected. e.g. "...these components are part of the well-established 'positive' and 'negative' transcriptional-translational feedback loops...cyc and cyc (positive) and per and tim (negative)".

Lines 293-295. This description of the clock is not strictly correct. Conceptually, there are in fact three loops in the transcription-translational feedback loop mechanism that comprises the molecular clock. I suggest the authors change the text accordingly. A correct description would be as follows:

"...consisting of three loops: the positive loop consists of activators clk/bmal1 in mammals and clk/cyc in fruit flies, the negative loop consists of repressors per/cry in mammals and per/tim in fruit flies, and a third, interlocking loop is recognized as the combination of reverb/ror or pdp/vri in mammals and flies, respectively".

Reviewer #2 (Remarks to the Author):

The manuscript by Zhang and colleagues present a comprehensive study underlying the ability of the nocturnal moth *Spodoptera litura*, a noctuid pest with devastating impacts on agriculture in Asia, to rhythmically detoxify plant secondary metabolites and xenobiotics, including pesticides. They first show that *S. litura* larvae exhibit diurnal rhythmic behavior in locomotion and feeding that peaks during the night, consistent with their nocturnal lifestyle, as well as rhythms in defecation, and all of those persist to some extent in constant darkness conditions, thus revealing regulation by the circadian system. Using RNA-seq and quantitative RT-PCR, the authors went on to identify and characterize the rhythmicity of several core clock genes in both LD and DD conditions, as well as of a set of 20 genes encoding detoxification enzymes (P450s, GSTs, COEs) expressed in the larval gut and fat body that are peaking during the day. They further enhanced their study by showing that RNAi-mediated knockdown of four of the core circadian clock genes all together (Clk, Cyc, Per and Tim) consistently and robustly decreased expression levels of a representative pool of the detoxifying enzymes, while knocking down the clock repressor Cwo enhanced it, demonstrating that genes encoding detoxification enzymes are regulated by the clock (and most likely direct clock targets). Importantly, the authors also show that knocking down Clk, Cyc, Per and Tim together resulted in increased sensitivity to the pesticide imidacloprid, and that its application on larvae affected larvae significantly more at night than during the day, thereby demonstrating an adaptive function of the clock in pesticide detoxification. They also annotated e-boxes in the regulatory regions of detoxification genes and demonstrated in vitro that their

function in circadian transcription through CLK:CYC/BMAL1 binding is conserved with *Drosophila*.

The study presented in this manuscript is comprehensive, the experimental work is of high quality and well controlled, and the conclusions supported by experimental results. The work presents very interesting findings with respect to clock control of pesticide detoxification that will be of broad interest for the clock community and may have broad implications for the control of *S. litura*, a pest present in Asia and whose range is currently expanding around the globe. In my opinion, the work is of high caliber, and *Communications Biology* would be a good home for it. I only have minor comments related to some of the writing and interpretation to improve the readability of the manuscript.

1. The authors argue lines 132-134 that the defecation rhythm in DD is likely not directly controlled endogenously by the clock, but rather just a mere consequence of the feeding rhythm. The rationale for this argument is not clear to me given that the amplitudes of rhythms in food consumption and defecation in DD (shown in Fig1) are quite similar. I do not think the authors' interpretation can be made.

2. Lines 189-191, 199-200: The authors claim that the 9 core clock genes exhibit "clear and regular 24-hr patterns in the head". Based on the results shown in Fig 3 and supplementary Fig 4, this does not appear to be true for *Clk*, *Cyc* and *pdp1* (neither by RNA-seq or qPCR) that look arrhythmically expressed, just like what has been described previously in another lepidopteran (Lugena et al, 2019, Genome-wide discovery of the daily transcriptome, DNA regulatory elements and transcription factor occupancy in the monarch butterfly brain, *PLoS Genet*, 15(7):e1008265). I think the sentence line 189-191 should be revisited. The same applies to the one lines 199-200.

3. Lines 211-212: The rationale presented by the authors for mixing siRNAs targeting the two circadian activators *Clk* and *Cyc* with those of repressors *tim* and *per* is not stated clearly. Was it not simply to ascertain that the clock would be rendered dysfunctional? Based on what is known in *Drosophila* and the monarch butterfly, one would expect that a knockdown of *Clk* and *Cyc* would be sufficient to abrogate activation.

3. Do the authors know whether clock genes are endogenously expressed in the *S. litura* cell line used for luciferase assays in Fig 6? It seems to me that clock genes are endogenously expressed, as activation is observed in absence of *Clk* or *Cyc* overexpression, and can be repressed by overexpression of *Per* or *Cwo* alone. I think that it would be worth discussing.

4. This is just a suggestion, but maybe moving the discussion on "similarity and differences of transcriptional rhythmicity of clock genes..." before the one on E-boxes would increase its flow.

5. Line 251-252: I do not understand what the last part of the sentence "and canonical E-boxes appeared more frequently in this region (Fig 5b)" is referring to, as Fig 5b only shows the number of detoxification genes relative to the distance between the E-box and the TSS. The authors should clarify this or remove the part of the sentence after the comma.

6. Line 55: It is unclear to me how "... the throughput of food in the alimentary tract" explains why night-time feeding determines the exposure of plant metabolites. Please, clarify.

7. Lines 65-67: Due to grammatical issues, this sentence is unclear. Please rewrite.

8. Line 77: The list of clock outputs listed includes both physiology and behavior. I suggest adding "and behavior".

9. Lines 79-80: In the same vein, feeding is a behavior but digestion is a physiological process here. Please correct.

10. Lines 85-87. It seems that this sentence is misplaced. The flow of this paragraph would be increased if this sentence were moved to line 105.

11. Line 93: The term "eventually" lacks precision. The inhibition happens every 24 hrs. Please,

correct.

12. Line 108: The authors should specify that the rhythms are from larvae here.

13. Line 123: What do the authors mean by "antiphase"? Trough? If so, clarify.

14. Line 127: What does a "throughput time" means? Please clarify.

15. Lines 135-137: It seems that the authors reversed the phase-advanced/delay and shorter/longer period.

16. Why would be a nocturnal feeder would make this insect a "formidable polyphagous feeder"? It seems to me that polyphagy may have more to do with how diverse a diet one can digest rather than time of feeding or intense digestion. Please clarify or remove the second part of the sentence following the coma.

17. Line 159: Please define P450, GST, COE and APN the first time they appear.

18. Line 160: Based on the method section, the authors have identified genes, not transcripts (i.e. splice variants). "transcripts" should thus be removed here and changed by "gene" wherever appropriate in the manuscript.

19. Line 169: The reference to Supplementary Table 4 must be a mistake. This table correspond to annotation of E-boxes, not expression levels of detoxification genes. Please, correct.

20. Line 183: "Cyc" should be replaced throughout the manuscript by Bmal1. While I understand that the nomenclature for this gene in insect is a bit confusing, it should be named Bmal1 in lepidopteran as lepidopteran orthologues of mammalian Bmal1/Drosophila Cyc have retained the C-terminal domain found on mammalian Bmal1 (unlike in Drosophila in which it has been lost).

21. Line 294: Clockwork mechanisms have been worked out in another lepidopteran, the monarch butterfly, and showed that they harbor a mammalian-like clock, with CLK:BMAL1 as circadian activators, and PER and CRY as repressors. Thus lepidopteran should be added here when referencing to mammalian components.

22. Lines 296-299: I do not think this sentence is necessary as it does not add any information relevant to the discussion. In fact, it is even a bit distractive.

23. Line 299: Instead of "drivers that adjust", I would recommend the authors use "cues that drive".

24. Line 338: How do the authors know that daytime activated detoxification genes were within the same chromatin loop? As they did not demonstrate this experimentally, this should only be a suggestion. Please tone down this statement.

25. Line 355: Clk and Cyc expression in larval heads looks arrhythmic rather than biphasic to me. Have the author performed any statistical analysis of 12-hr rhythms to back up this claim?

Response to the reviewers' comments

Reviewer #1:

1. Line 108. Grammar. 'Strongly' does not work well in this sentence. Perhaps 'robust' would be a better choice?

Reply: We accepted this comment and replaced "strongly" with "robust" (now Line 112).

2. Line 152. Insert 'a' between 'it' and 'most'.

Reply: We accepted this comment but according to the comment 16 of Reviewer #2 we deleted this part of the sentence (now Lines 158-159).

3. Line 213. The authors have a misconception regarding the organization of the molecular clock. Clk and cyc form the positive loop, and per/tim the negative loop. As such this sentence should be corrected. e.g. "...these components are part of the well-established 'positive' and 'negative' transcriptional-translational feedback loops...cyc and cyc (positive) and per and tim (negative)".

Reply: We corrected the description according to the reviewer's comment (now Lines 226-227).

4. Lines 293-295. This description of the clock is not strictly correct. Conceptually, there are in fact three loops in the transcription-translational feedback loop mechanism that comprises the molecular clock. I suggest the authors change the text accordingly. A correct description would be as follows: "...consisting of three loops: the positive loop consists of activators clk/bmal1 in mammals and clk/cyc in fruit flies, the negative loop consists of repressors per/cry in mammals and per/tim in fruit flies, and a third, interlocking loop is recognized as the combination of reverb/ror or pdp/vri in mammals and flies, respectively".

Reply: We changed the description according to the reviewer's comment (now Lines 311-315).

Reviewer #2:

1. The authors argue lines 132-134 that the defecation rhythm in DD is likely not directly controlled endogenously by the clock, but rather just a mere consequence of the feeding rhythm. The rationale for this argument is not clear to me given that the amplitudes of rhythms in food consumption and defecation in DD (shown in Fig1) are quite similar. I do not think the authors' interpretation can be made.

Reply: We agree with the reviewer's interpretation that the evidence does not support the original wording and deleted this sentence. (now Lines 138-140)

2. Lines 189-191, 199-200: The authors claim that the 9 core clock genes exhibit "clear and regular 24-hr patterns in the head". Based on the results

shown in Fig 3 and supplementary Fig 4, this does not appear to be true for Clk, Cyc and pdp1 (neither by RNA-seq or qPCR) that look arrhythmically expressed, just like what has been described previously in another lepidopteran (Lugena et al, 2019, Genome-wide discovery of the daily transcriptome, DNA regulatory elements and transcription factor occupancy in the monarch butterfly brain, PLoS Genet, 15(7):e1008265). I think the sentence line 189-191 should be revisited. The same applies to the one lines 199-200.

Reply: We rewrote these two sentences to make the descriptions more rigorous and precise in agreement with the reviewer's comments (now Lines 201-202 and Line 212).

3. Lines 211-212: The rationale presented by the authors for mixing siRNAs targeting the two circadian activators Clk and Cyc with those of repressors tim and per is not stated clearly. Was it not simply to ascertain that the clock would be rendered dysfunctional? Based on what is known in *Drosophila* and the monarch butterfly, one would expect that a knockdown of Clk and Cyc would be sufficient to abrogate activation.

Reply: As the reviewer suggests, we aimed to disrupt the core circadian system in *S. litura* to see how this affected the expression of detoxification genes. Therefore, we now only present our knockdown results targeting *Clk* and *Bmal1*, (renamed *Bmal1* based on the reviewer's comment 20) (Fig. 4). As the new results showed, similar to *Drosophila* and the monarch butterfly, a knockdown of *Clk* and *Bmal1* was sufficient and much simpler to explain the regulation between core circadian genes and the detoxification process. (now Lines 221-228)

4. This is just a suggestion, but maybe moving the discussion on "similarity and differences of transcriptional rhythmicity of clock genes..." before the one on E-boxes would increase its flow.

Reply: We followed this suggestion and moved that section to just before the section "Clock-controlled E-box binding..." (now Lines 339-363).

5. Line 251-252: I do not understand what the last part of the sentence "and canonical E-boxes appeared more frequently in this region (Fig 5b)" is referring to, as Fig 5b only shows the number of detoxification genes relative to the distance between the E-box and the TSS. The authors should clarify this or remove the part of the sentence after the comma.

Reply: We accepted the reviewer's comment and removed the part of the sentence after the comma (now Lines 266-267).

6. Line 55: It is unclear to me how "... the throughput of food in the alimentary tract" explains why night-time feeding determines the exposure of plant metabolites. Please, clarify.

Reply: Based on the reviewer's concern, we changed this statement by deleting

the part of the sentence referring to “throughput of food in the alimentary tract” to “Night feeding also determines the exposure time to plant secondary metabolites and xenobiotics such as insecticides.”. (now Lines 54-56)

7. Lines 65-67: Due to grammatical issues, this sentence is unclear. Please rewrite.

Reply: We rewrote this sentence to simplify and clarify it as a concluding statement for the paragraph, as follows: “These properties, along with its high fecundity, make *S. litura* one of the most difficult pests to control.” (now Lines 66-68).

8. Line 77: The list of clock outputs listed includes both physiology and behavior. I suggest adding “and behavior”.

Reply: We accepted this comment and made additional minor changes in accordance with this and some of the reviewer’s other suggestions (see below). (now Lines 77-78)

9. Lines 79-80: In the same vein, feeding is a behavior but digestion is a physiological process here. Please correct.

Reply: We corrected this as the reviewer suggested (now Lines 80-81).

10. Lines 85-87: It seems that this sentence is misplaced. The flow of this paragraph would be increased if this sentence were moved to line 105.

Reply: We accepted this comment and moved this sentence to Lines 105-107 to improve the flow. (now Lines 107-109)

11. Line 93: The term “eventually” lacks precision. The inhibition happens every 24 hrs. Please, correct.

Reply: Thank you for the correction. We deleted the word “eventually” to make the statement more precise. (now Line 95)

12. Line 108: The authors should specify that the rhythms are from larvae here.

Reply: Although the original sentence referred to larvae and the last larval instars, we accepted this comment and revised the sentence by adding more information about the larval stages when the rhythms occur (now Line 110 and Lines 112-116).

13. Line 123: What do the authors mean by “antiphase”? Trough? If so, clarify.

Reply: We replaced “antiphase” with “trough” to make the meaning of the sentence clearer (now Line 129).

14. Line 127: What does a “throughput time” means? Please clarify.

Reply: We changed the description to “peaking at ZT9 during daytime in a 24 hour cycle” to make it clearer (now Line 133).

15. Lines 135-137: It seems that the authors reversed the phase-advanced/delay and shorter/longer period.

Reply: We agreed with this comment and accordingly changed the wording to “The free-running rhythm in locomotion (Fig. 1a) seemed phase-delayed relative to the entrained rhythm, suggesting a circadian period, τ , longer than 24 h, while that in feeding (Fig. 1b) was phase advanced, suggesting a τ shorter than 24 h.” (now Lines 141-143).

16. Why would be a nocturnal feeder would make this insect a “formidable polyphagous feeder”? It seems to me that polyphagy may have more to do with how diverse a diet one can digest rather than time of feeding or intense digestion. Please clarify or remove the second part of the sentence following the coma.

Reply: We accepted this comment and deleted this part of the sentence (now Lines 158-159).

17. Line 159: Please define P450, GST, COE and APN the first time they appear.

Reply: We accepted this comment and defined these terms the first time they appear in the text (now Lines 163-167).

18. Line 160: Based on the method section, the authors have identified genes, not transcripts (i.e. splice variants). “transcripts” should thus be removed here and changed by “gene” wherever appropriate in the manuscript.

Reply: We have accepted this comment and used the appropriate terms throughout the manuscript (e.g., Line 117 and Line 172).

19. Line 169: The reference to Supplementary Table 4 must be a mistake. This table correspond to annotation of E-boxes, not expression levels of detoxification genes. Please, correct.

Reply: We accepted the reviewer’s comment and deleted the reference to Supplementary Table 4. We actually used data from Fig. 2b (referred to in the previous sentence) to estimate the number of detoxification genes more highly expressed in daytime (now Line 180).

20. Line 183: “Cyc” should be replaced throughout the manuscript by Bmal1. While I understand that the nomenclature for this gene in insect is a bit confusing, it should be named Bmal1 in lepidopterans as lepidopteran orthologues of mammalian Bmal1/Drosophila Cyc have retained the C-terminal domain found on mammalian Bmal1 (unlike in Drosophila in which it has been lost).

Reply: We followed the reviewer’s suggestion about replacing “Cyc” with

“*Bmal1*” after we checked the sequence homology of mammalian *Bmal1* in *S.litura*. The sequence alignment analysis showed that the homology of the *S.litura* is higher with mammalian *Bmal1* particularly in the C-terminal domain than with *Drosophila Cyc*. We also used “*Cyc*” when the sentences described the results in *Drosophila* and added two notes about this in Lines 90 and 194.

21. Line 294: Clockwork mechanisms have been worked out in another lepidopteran, the monarch butterfly, and showed that they harbor a mammalian-like clock, with CLK:BMA1 as circadian activators, and PER and CRY as repressors. Thus lepidopterans should be added here when referencing to mammalian components.

Reply: We added the paper suggested (ref. 39) which uncovered the clockwork mechanism in another lepidopteran, the monarch butterfly, as the reviewer suggested (now Lines 312-315).

22. Lines 296-299: I do not think this sentence is necessary as it does not add any information relevant to the discussion. In fact, it is even a bit distractive.

Reply: We deleted these lines according to the reviewer’s suggestion (now Lines 316-318).

23. Line 299: Instead of “drivers that adjust”, I would recommend the authors use “cues that drive”.

Reply: We accepted this comment and changed the wording as the reviewer suggested (now Lines 318-319).

24. Line 338: How do the authors know that daytime activated detoxification genes were within the same chromatin loop? As they did not demonstrate this experimentally, this should only be a suggestion. Please tone down this statement.

Reply: We agreed with this comment and changed the wording to be more hypothetical. We also added a reference to the *S. litura* genome paper (ref. 7, Line 381) which supports the idea that many of the detoxification genes under consideration are close enough to each other within their gene clusters to be within the same chromatin loop(s) (now Lines 381-383).

25. Line 355: *Clk* and *Cyc* expression in larval heads looks arrhythmic rather than biphasic to me. Have the author performed any statistical analysis of 12-hr rhythms to back up this claim?

Reply: We performed statistical analysis of 12 h rhythms in *Clk* and *Cyc* expression in triplicate using the nonparametric test, JTK_CYCLE, which supported the idea that *Clk* and *Cyc* are expressed with a biphasic pattern. Please see Methods section “Statistics” for more information (now Lines 582-584).

REVIEWERS' COMMENTS:

Reviewer #2 (Remarks to the Author):

The authors have addressed most of my comments reasonably well, with a few exceptions. Some of the authors' responses raise additional minor points that, once addressed, will tighten up the manuscript and make it ready for publication.

1. The sentence line 201-202 that the authors modified in response to comment #2 in my original review is confusing. The results presented by the authors in Fig. 3 and Supp. Fig. 4 argue that out of 9 annotated core clock genes, only 6 (*per*, *tim*, *cwo*, *cry1*, *cry2* and *vri*) display robust 24-hr rhythms in the head in LD, and the expression of those is not biphasic (i.e. there is only one peak over a 24-hr period). *Clk* and *bmal1* (and perhaps *pdp1*) look biphasic. The way the sentence reads right now makes it sound like the 6 rhythmic genes (*per*, *tim*, *cwo*, *cry1*, *cry2* and *vri*) show biphasic profiles. I do not think this is what the authors want to say and they should reword this sentence to make it clear. Also, line 212, the use of "semi circadian" for 12-hr rhythms is odd (as circadian is conventionally used for 24-hr rhythms). It would be more appropriate to simply term them 12-hr rhythms.

2. With respect to changing *Cyc* by *Bmal1*, the authors have done so appropriately throughout the manuscript, but not in Supp. Fig 4, where it should also be changed. One exception is line 194 where the authors refer to *Bmal1* as "also called *Cyc*-like". This is incorrect. The only instance to my knowledge in which *Bmal1* was called *Cyc*-like was in Zhang et al. (2017, PNAS, E7516-E7525) and it was used to describe a *Bmal1* mutant lacking the sequence corresponding to the C-terminal domain that has been lost in *Drosophila Cyc* (hence, the name *Cyc*-like). I suggest that the authors simply remove this statement.

3. Regarding the description of the clockwork mechanism and its comparison between insects and mammals in the first paragraph of the discussion, I strongly disagree with some of the changes made in response to the other reviewer. Specifically, unlike what was suggested to/written by the authors, the core clockwork mechanism is not composed of three loops, but TWO interlocked ones. The core feedback loop is composed of two arms: a positive arm in which CLK:BMAL1 in mammals and monarch butterfly, and CLK:CYC in *Drosophila*, activate transcription of target genes, including that of negative elements (*per/cry*), and a negative arm in which the negative elements (PER/CRY in mammals and monarch, and PER/TIM in *Drosophila*) repress their own CLK:BMAL1-mediated transcription and those of other target genes. The second loop is a stabilizing loop, and relies on *Reverb/Ror* in mammals and *Pdp1/Vri* in insects. The description lines 311-315 needs to be modified to reflect the proper current state of knowledge in the field. In addition, the authors should cite seminal papers that show that in monarchs CLK:BMAL1 act as circadian activators (Zhang et al., Vertebrate-like CRYPTOCHROME 2 from monarch regulates circadian transcription via independent repression of CLOCK and BMAL1 activity, 2017, PNAS, E7516-E7525) and that repression is ensured by PER/CRY (Zhu et al, The two CRYs of the butterfly, *Curr Biol.* 2005 15(23):R953-4; Zhu et al., Cryptochromes define a novel circadian clock mechanism in monarch butterflies that may underlie sun compass navigation, *PLoS Biol.* 2008 6(1):e4).

4. The authors responded that 12-hr rhythms of *Clk* and *Bmal1* are supported by statistical analysis. However, there are not enough details in the method section about this analysis to assess its validity. Which parameters/settings did the authors use in *JTK_CYCLE* to assess 12-hr rhythms? And which ones did they use to assess 24-hr rhythms? There should be some differences between the two, and this should be clearly stated in the method section.

5. The last sentence of the discussion (lines 423-427) is complicated and difficult to follow, as it contains too many ideas. I suggest that the authors break it in two sentences.

Response to reviewer comments

1. The sentence line 201-202 that the authors modified in response to comment #2 in my original review is confusing. The results presented by the authors in Fig. 3 and Supp. Fig. 4 argue that out of 9 annotated core clock genes, only 6 (*per*, *tim*, *cwo*, *cry1*, *cry2* and *vri*) display robust 24-hr rhythms in the head in LD, and the expression of those is not biphasic (i.e. there is only one peak over a 24-hr period). *Clk* and *bmal1* (and perhaps *pdp1*) look biphasic. The way the sentence reads right now makes it sound like the 6 rhythmic genes (*per*, *tim*, *cwo*, *cry1*, *cry2* and *vri*) show biphasic profiles. I do not think this is what the authors want to say and they should reword this sentence to make it clear. Also, line 212, the use of “semi circadian” for 12-hr rhythms is odd (as circadian is conventionally used for 24-hr rhythms). It would be more appropriate to simply term them 12-hr rhythms.

Reply: We accepted the reviewer’s comment and rewrote these lines to make them clearer. (now Lines 192-196, Line 205)

2. With respect to changing *Cyc* by *Bmal1*, the authors have done so appropriately throughout the manuscript, but not in Supp. Fig 4, where it should also be changed. One exception is line 194 where the authors refer to *Bmal1* as “also called *Cyc*-like”. This is incorrect. The only instance to my knowledge in which *Bmal1* was called *Cyc*-like was in Zhang et al. (2017, PNAS, E7516-E7525) and it was used to describe a *Bmal1* mutant lacking the sequence corresponding to the C-terminal domain that has been lost in *Drosophila Cyc* (hence, the name *Cyc*-like). I suggest that the authors simply remove this statement.

Reply: We updated Supplementary Fig. 4 to change “*SlituCyc*” to “*SlituBmal1*” (Supplementary Fig. 5 as well). We removed the statement “also called *Cyc*-like” as the reviewer suggested. (now Line 186)

3. Regarding the description of the clockwork mechanism and its comparison between insects and mammals in the first paragraph of the discussion, I strongly disagree with some of the changes made in response to the other reviewer. Specifically, unlike what was suggested to/written by the authors, the core clockwork mechanism is not composed of three loops, but TWO interlocked ones. The core feedback loop is composed of two arms: a positive arm in which CLK:BMAL1 in mammals and monarch butterfly, and CLK:CYC in *Drosophila*, activate transcription of target genes, including that of negative elements (*per/cry*), and a negative arm in which the negative elements (PER/CRY in mammals and monarch, and PER/TIM in *Drosophila*) repress their own CLK:BMAL1-mediated transcription and those of other target genes. The second loop is a stabilizing loop, and relies on *Reverb/Ror* in mammals and *Pdp1/Vri* in insects. The description lines 311-315 needs to be modified to reflect the proper current state of knowledge in the field. In addition, the

authors should cite seminal papers that show that in monarchs CLK:BMAL1 act as circadian activators (Zhang et al., Vertebrate-like CRYPTOCHROME 2 from monarch regulates circadian transcription via independent repression of CLOCK and BMAL1 activity, 2017, PNAS, E7516-E7525) and that repression is ensured by PER/CRY (Zhu et al., The two CRYs of the butterfly, Curr Biol. 2005 15(23):R953-4; Zhu et al., Cryptochromes define a novel circadian clock mechanism in monarch butterflies that may underlie sun compass navigation, PLoS Biol. 2008 6(1):e4).

Reply: We accepted this comment and rewrote these sentences as the reviewer suggested. We also added to the references the recommended papers which report the clockwork mechanisms in monarch butterfly. (now Lines 296-302)

4. The authors responded that 12-hr rhythms of Clk and Bmal1 are supported by statistical analysis. However, there are not enough details in the method section about this analysis to assess its validity. Which parameters/settings did the authors use in JTK_CYCLE to assess 12-hr rhythms? And which ones did they use to assess 24-hr rhythms? There should be some differences between the two, and this should be clearly stated in the method section.

Reply: JTK_CYCLE was used to measure period length without specific parameters (described in Lines 559-560: "For analysis of rhythmic expression patterns, the nonparametric test, JTK_CYCLE, was used to.....", but we controlled the window within 12-24 h based on our samples. Additionally, we further confirmed the 12-hour rhythms in the 72 h expression profiles of *SlituClk* and *SlituBmal1* (Fig. 3). These data are presented as 12-hr rhythms according to the time point of peaks that appeared under LD.

5. The last sentence of the discussion (lines 423-427) is complicated and difficult to follow, as it contains too many ideas. I suggest that the authors break it in two sentences.

Reply: We split up the ideas and rewrote the description with two sentences to make them easier to follow. (now Lines 402-408)